# Mapping the effective coverage of modern contraceptive services in Ethiopia

**Samrawit Birhanu Alemu**[1]*, **Molla Yigzaw Birhanu**[1], **Mekdes Tamiru Yizengaw**[2], **Melaku Birhanu Alemu**[3,4]

1 Department of Public Health, Debre Markos University, Debre Markos, Ethiopia, 2 St. Lideta Health Science College, Addis Ababa, Ethiopia, 3 Department of Health Systems and Policy, Institute of Public Health, University of Gondar, Gondar, Ethiopia, 4 Curtin School of Population Health, Curtin University, Perth, Western Australia, Australia

* samribtm@gmail.com

## Abstract

### Introduction

Modern contraceptive services are vital for reducing maternal and infant morbidity and mortality. However, in Ethiopia, the effective coverage (quality-adjusted coverage) of these services remains low and varies significantly across administrative regions. Despite these disparities, the spatial distribution of effective coverage is not well understood, limiting the ability to implement targeted interventions in areas with low coverage.

### Methods

We used the 2019 Ethiopia Demographic Health Survey (EMDHS) to estimate the crude coverage of modern contraceptive services. In addition, we used the 2021–2022 Ethiopia Services Provision Assessment (ESPA) to estimate the quality of modern contraceptive services provision. The two datasets were linked using the Euclidean buffer link method to estimate the effective coverage. The effective coverage was calculated as the product of coverage and quality estimates for each health facility. The geospatial estimates were presented using the coordinates from the ESPA survey. We analysed the spatial distribution of modern contraceptive coverage using ArcGIS 10.7 software. The Global Moran's I statistic was used to assess the spatial autocorrelation, while the Getis-Ord Gi* statistic was used to identify high and low clusters. The Kriging interpolation method was used to estimate effective coverage in unsampled areas based on sampled clusters.

### Results

The effective coverage of modern contraceptive services in Ethiopia was 20% (95% confidence interval (CI): 19.87, 21.82), which was notably lower than the crude

**Data availability statement:** The data underlying the results presented in the study are available from the demographic health survey (DHS) program (https://dhsprogram.com/).

**Funding:** The author(s) received no specific funding for this work.

**Competing interests:** The authors declare that they have no competing interests.

coverage of 30% (95% CI: 28.93, 31.62), resulting in a gap of 10% points. The spatial distribution analysis showed that effective modern contraceptive coverage had a significant spatial autocorrelation in Ethiopia, with the Global Moran's index value of 1.92 (p-value < 0.001). The effective coverage of modern contraceptive services was notably higher in Central Amhara, Western Amhara, Central Oromia, the western part of South West Ethiopia, Sidama, and the northern part of the Southern Nations, Nationalities, and Peoples (SNNP). Addis Ababa, western Amhara, northeast Benishangul-Gumuz, eastern part of South West Ethiopia, north SNNP, Sidama, and northern and central Oromia had hotspots for effective coverage of modern contraceptives. On the other hand, Afar, western Gambela, western Oromia, western Benishangul-Gumuz, Dire Dawa, Harari and Somali had cold spots for effective coverage of modern contraceptive services.

## Conclusions

There is a significant spatial variation in the effective coverage of modern contraceptive services in Ethiopia. Afar, western Gambela, western Oromia, western Benishangul-Gumuz, Dire Dawa, Harari, and Somali had low effective coverage. To enhance equitable access, policymakers should prioritise interventions such as improving healthcare infrastructure and strengthening service delivery to improve modern contraceptive coverage and quality in underserved areas.

## Introduction

Family planning is widely recognised as a critical measure in reducing maternal and infant mortality and morbidity, contributing significantly to the achievement of national and global health objectives [1]. In 1994, the International Conference on Population and Development (ICPD) Programme of Action (PoA) emphasises that family planning is not just a demographic tool but a fundamental right that enhances maternal and child health [2]. It helps reduce maternal mortality by preventing unintended pregnancies, improves child health through optimal birth spacing, empowers women with reproductive choices, and promotes gender equality by supporting education and economic opportunities [2,3]. Family planning methods are generally classified into two categories: traditional methods, including the rhythm method and withdrawal, and modern methods. Modern options include hormonal methods (such as implants, oral contraceptive pills, and injectables), barrier methods (such as condoms and diaphragms), long-acting reversible contraceptives (such as intrauterine devices), permanent methods (such as sterilisation), and emergency contraception [4–7].

Globally, 77% of the 1.2 billion women of reproductive age (15–49) needing family planning were using modern contraceptives in 2022, leaving approximately 270 million women with unmet need [8]. Regional inequalities persist, with Sub-Saharan Africa bearing the greatest burden. Modern contraception

remains inaccessible to 37% of women in the region [8]. According to the 2019 Ethiopia Mini Demographic and Health Survey (EMDHS), the national contraceptive prevalence for modern methods is around 28.1%, with significant regional variation. In Addis Ababa, coverage is higher at 47.6%, while in Somali, it is much lower, at about 3.4% [9].

Proper birth spacing through contraception prevents approximately 1.9 million child deaths each year [10]. Improved family planning access not only enhances maternal health but also reduces child mortality, potentially preventing 70,000 maternal deaths each year [11,12]. Moreover, effective family planning programs contribute to poverty alleviation and food security in countries with rapid population growth by optimizing demographic conditions [13,14]

Quality provision of health services is the primary predictor of achieving sustainable development goals [15]. Quality of care is a complex concept that requires a wide range of information to be measured, including facility readiness, processes of care and users' experiences, and effects of care, to conceptualise quality from inputs, processes, and outcome perspectives, the Donabedian model of quality health care is widely used [16].

Effective coverage of modern contraceptive services reflects services utilisation (crude coverage) and care quality. In LMICs, significant gaps exist between these measures, highlighting the need for quality improvements. Studies show that in 33 countries, while median crude coverage was 76.6%, quality-adjusted coverage dropped to 52.9%, revealing a 23.7 percentage point gap [17]. In sub-Saharan Africa, where unmet contraceptive needs remain high, these disparities are concerning. For instance, in Kenya, crude family planning coverage was 57.9%, but quality-adjusted coverage fell to 27.3% [18].

The quality of healthcare services remains a critical concern in achieving effective coverage in Ethiopia [19,20]. Poor-quality health services, such as inadequate counselling, insufficient provider knowledge, and a lack of privacy during consultations, directly affect the utilisation of modern contraceptive methods [21,22]. Maternal mortality rates that continue to remain high may be caused by inadequate care, which was not reflected in the basic coverage measures [23]. Low service quality results in poor effective coverage and reduces the potential health benefits from using these services, counteracting the relatively high crude coverage rates for modern contraceptive services [24]. Crude coverage cannot assess the degree of adherence to care standards or how services were delivered. It only evaluates access to services without considering the quality of care [25,26].

Despite significant growth in the number of health facilities delivering family planning, the quality of these services remains persistently inadequate in Ethiopia [24]. Bridging the gap between crude and quality-adjusted coverage requires targeted interventions [17]. Strengthening the quality of contraceptive services will enhance reproductive health outcomes and contribute to broader health equity and development goals in Ethiopia [17,18,24]. However, the quality-adjusted coverage of modern contraceptive services and their distribution have not been studied. Therefore, this study aims to assess the spatial distribution of effective coverage of modern contraceptive services in Ethiopia by integrating coverage and quality dimensions. The findings will provide critical insights for policymakers to enhance service quality and expand quality-adjusted contraceptive coverage, ultimately improving reproductive health outcomes in Ethiopia.

## Methods

### Study area

This study was conducted in Ethiopia, a federal state in the Horn of Africa. The country has a diverse population and geographical landscape, with significant variations in healthcare infrastructure and service availability between urban and rural areas. The country has a decentralised, three-tiered healthcare system: primary (health posts, centres, and primary hospitals), secondary (general hospitals), and tertiary (specialised and referral hospitals) [27]. Ethiopia's healthcare system focuses on expanding access to essential health services, including family planning, through its Health Extension Program, which deploys trained health workers to rural and underserved areas [28].

## Study design and data source

This study used a cross-sectional design based on secondary data from the EMDHS 2019 and the ESPA 2021–2022. The EMDHS is a nationally representative, comprehensive dataset that assesses the demographic and health status of Ethiopia. The ESPA is a facility assessment that gathers information on service availability and quality of care. Both datasets are publicly available upon request from the Demographic and Health Survey (DHS) at https://dhsprogram.com.

## Sampling technique

The 2019 EMDHS used the 2019 Ethiopia Population and Housing Census (EPHC) as its sampling frame, which included 149,093 enumeration areas (EAs), each averaging 131 households. The survey followed a two-stage stratified sampling approach, selecting 305 EAs proportionally across urban and rural areas. A household listing was conducted, and 30 households per cluster were systematically chosen. Women aged 15–49 were interviewed, and height and weight measurements were collected from children under five. 8,885 of the 9,012 women who were eligible for individual interviews in the interviewed households finished the interview [9].

The 2021–2022 ESPA used a stratified random sample of 1,407 health facilities. All 413 hospitals were included due to their crucial role in the health system. A representative sample of 310 health centres, 328 health posts, and 356 clinics was selected, with oversampling of key facilities to improve survey precision. Data collection covered 1,158 facilities (82% of the sample), with some exclusions due to closures, security concerns, or facility conversions. Most facilities were health posts (65%), primarily government-managed (83%), and located in rural areas (77%), with the highest concentration in Oromia, Amhara, and SNNP regions [29].

## Data quality control

Based on the standard DHS questionnaires used by the DHS Program, which were modified to account for the Ethiopian population and health concerns, the DHS health survey employs a carefully crafted questionnaire. Before the actual survey, the data collection staff received training and pretesting. Furthermore, quality controllers provided assistance and oversight during the data collection process [9,29]. Fieldwork supervision was conducted by the Ethiopian Public Health Institute (EPHI), the Ministry of Health (MoH), and ICF personnel, with regional coordinators monitoring data collection. Automated quality checks, field check tables, and double data entry were used to maintain data accuracy [9].

## Sample size

To assess contraceptive coverage, we analysed data from the individual record (IR) file of the EMDHS. Our initial sample consisted of 8,885 women of reproductive age at the time of the survey. However, we excluded 722 pregnant women from the dataset to ensure the accuracy of contraceptive coverage estimates. Furthermore, we removed data from the Tigray region for consistency in data analysis, as no corresponding records were available in the ESPA dataset. This resulted in excluding 733 observations from Tigray, leaving a final unweighted sample of 7,430 women (Fig 1).

To measure the quality of contraceptive services, we utilised health facility and family planning records from the ESPA dataset, which contains information from 1,158 health facilities nationwide. For our study, we specifically selected 648 health facilities that had available exit interview records related to family planning services (Fig. 1).

## Measure of variable

**Coverage variable.** The coverage measures in this study were derived from the current use of modern contraceptive services. Modern contraceptive services were coded as "yes" for those who use modern contraceptive methods and "no" for those who did not use any contraceptive and use traditional methods [30].

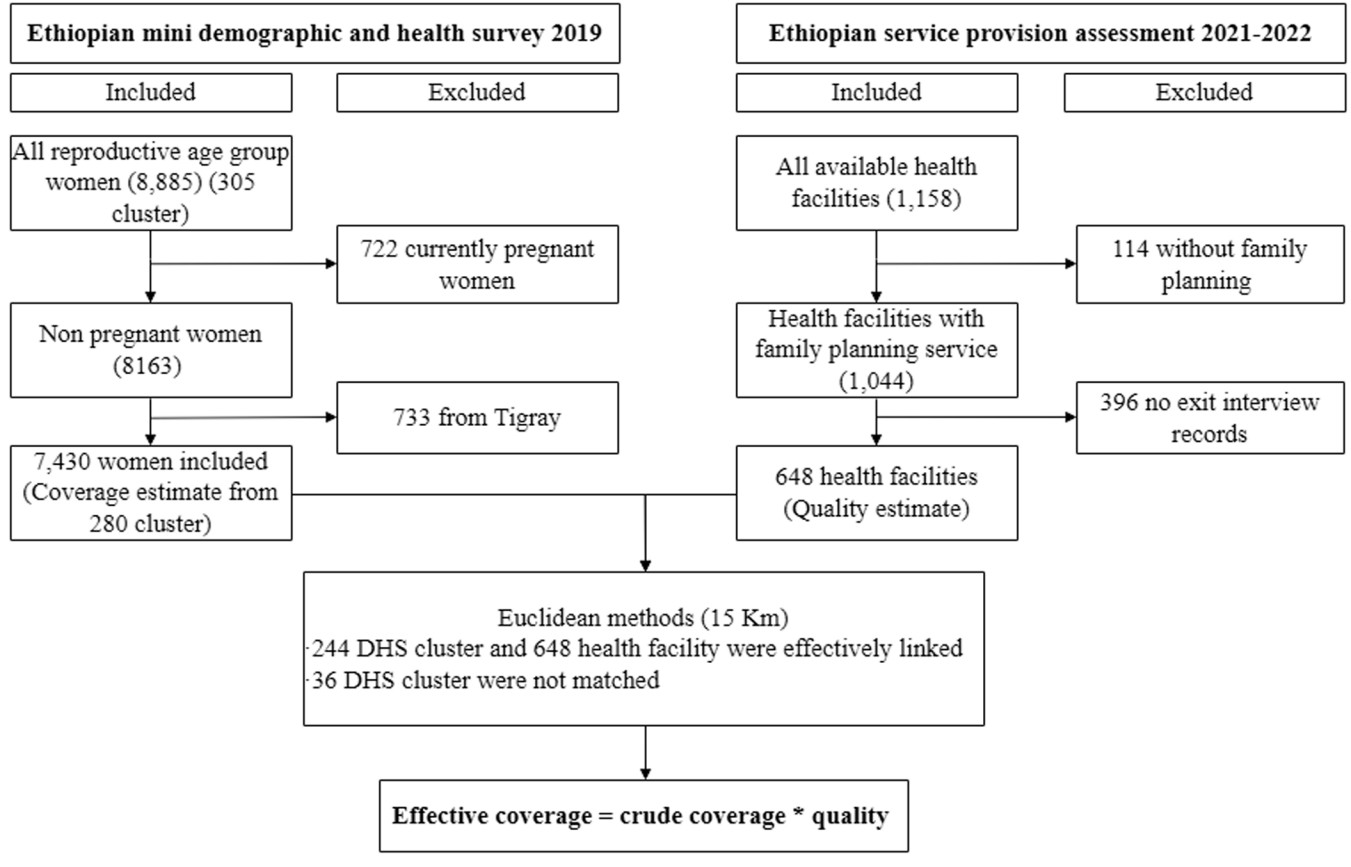

**Fig 1. Schematic illustration of women included in the study, EMDHS 2019 and health facilities included in the study, ESPA 2021–2022.**

## Quality of care variables

The quality measures were taken from the facility recode and family planning recode, ESPA 2021–2022. We use modern contraceptive recommended services, which were used for quality measures from the national guideline for family planning in Ethiopia [31]. The quality indicators were recoded as "yes" and "no" [24]. A total of 72 indicators were used, where 25 structural, 30 process, and 17 were outcome indicators (S1 Table).

## Data management and data analysis

The DHS statistics guide was used to clean up the data and handle any missing values. To restore the survey's representativeness, the data were weighted using the sample weights [32]. Stata 17 were used for analysis.

First, the crude coverage of modern contraceptive use was calculated for each cluster by determining the proportion of women of reproductive age who reported using any modern contraceptive method at the time of the survey. This was done by dividing the number of women using modern contraceptive services by the total number of women in each cluster.

Service quality was typically assessed across three key dimensions: structure, process, and outcome [16]. The quality of modern contraceptive services was calculated using additive weighting methods, which weight each quality component based on the number of quality indicators used [33–35]. Therefore, the structural component constituted 34.7% (25/72*100) of the total, process components had 41.7% (30/72*100), and the outcome component had 23.6%

(17/72*100). The score of each component was calculated as a proportion of recommended clinical actions done in family planning to the total number of indices in each component.

$$Structural\ component = \frac{Total\ number\ of\ "yes"\ responses\ in\ the\ componenent}{Total\ number\ of\ parameter\ in\ the\ structure\ component} * Weight$$

$$Process\ component = \frac{Total\ number\ of\ "yes"\ responses\ in\ the\ component}{Total\ number\ of\ parameter\ in\ the\ process\ component} * Weight$$

$$Outcome\ component = \frac{Total\ number\ of\ "yes"\ responses\ in\ the\ component}{Total\ number\ of\ parameter\ in\ the\ process\ component} * Weight$$

The EMDHS and ESPA datasets were linked using Euclidean (GPS-based) methods [36,37]. The linkage was conducted in STATA using the "geonear" command, which matched health facilities to the closest EMDHS survey cluster within a 15 km radius. If no cluster was found within this range, the health facility was linked to the nearest available cluster, even if it exceeded 15 km. In total, 244 DHS clusters and 648 health facilities were successfully linked. After linking the datasets, the effective coverage of modern contraceptive use was calculated by multiplying the quality of the health facility by the proportion of women using modern contraceptive services in each cluster [26,38,39]. Finally, the average effective coverage was calculated for each health facility to map the effective coverage.

Effective coverage = Quality of the health services $*$ crude coverage

The facility's characteristics, as well as the respondents' socioeconomic and demographic characteristics, were described using descriptive statistics. Lastly, tables and graphs were used to present the results. To account for methodological differences, an additional linkage approach was employed. Alongside the Euclidean (GPS-based) linkage, results were also reported using administrative linkage to provide a more comprehensive analysis (S2 Table).

## Spatial analysis

We use ArcGIS 10.7 software to analyse the spatial distribution after determining the effective coverage of modern contraceptive services. The geospatial estimates were presented using the coordinates from the ESPA survey. Sampling weights were applied using sample weights from DHS to adjust for population distribution and ensure representativeness.

## Spatial autocorrelation analysis

A measure of spatial autocorrelation called the Global Moran's I statistic was used to assess the spatial distribution of modern contraceptive services that work in Ethiopia. The distribution of modern contraceptive services across regions can be evaluated using this method to determine if it followed a spatial pattern or was random. Values close to −1 indicate dispersion, while values close to +1 indicate clustering. A Moran's I value of zero suggests a random distribution. A statistically significant Moran's I ($p < 0.05$) leads to rejection of the null hypothesis and indicates the presence of spatial autocorrelation [40].

## Hot spot analysis (Getis-Ord Gi* statistic)

A hot spot analysis was performed using the Getis-Ord Gi* statistic, which successfully identified noteworthy clusters of modern contraception that were high (hot places) and low (cold spots). To identify regions where modern contraceptive efforts were particularly successful or where coverage gaps were noticeable, this spatial statistical method is frequently used to find trends in geographic data. Z-scores and p-values were used to evaluate statistical significance, and 95% confidence intervals were used. A "hotspot" was denoted by a high Getis-Ord Gi* value, whereas a "cold spot" was denoted

by a low Gi* value. Clustering's statistical significance was evaluated using Z-scores, and significance levels were evaluated using p-values. To depict spatial differences, low-value clusters (cold spots) and high-value clusters (hotspots) were mapped [41].

### Spatial interpolation

The Kriging spatial interpolation approach was used to forecast the effective coverage of modern contraceptives in unsampled regions of Ethiopia based on sampled clusters [42]. Kriging is a geostatistical technique that uses spatial autocorrelation and data from neighbouring sampled points to estimate values at unmeasured places. In areas where direct data collection has not been carried out, this method guarantees more precise and trustworthy forecasts of the coverage of modern contraceptives [43].

### Ethics approval

The EMDHS dataset is available to the public upon request from the DHS website: https://dhsprogram.com/. We submitted a request to the DHS by briefly stating the objectives of this analysis and thereafter received permission to download the dataset.

## Results

A total of 7618 reproductive-age women were included. Of these, most (42.25%) were aged 15–24 years, followed by 25–35 years (36.01%) and 36–49 years (21.74%). Majority of the respondents (63.99%) were married and from rural areas (67.88%) (Table 1).
A total of 648 health facilities were included. Most of the facilities were hospitals (43.98%). Most of the facilities were governmental facilities (82.41%) (Table 2).

### Crude coverage and quality of modern contraceptive services

The crude coverage of modern contraceptive services in Ethiopia was 30% (95% CI: 28.93–31.62). The highest crude coverage was observed in SNNP at 36.1% and Amhara at 35.6%. In contrast, the lowest coverage was recorded in Somali at 8% and Afar at 17% (Table 5).

The modern contraception services had an overall quality score of 67% (95% CI: 66.04, 67.63). The quality varies from Addis Ababa's 73.7% to Gambela's 58.8%. The outcome component had the highest quality score, 90.24 (95% CI: 88.62, 91.86). The structural component had the second score, 66.6% (95% CI: 63.65, 69.49), ranging from 55.7% in Gambela to 77.5% in Addis Ababa. Lastly, the process component had a 53.77% (95% CI: 52.77, 54.77) quality score, ranging from 48% in Afar to 60.7% in Addis Ababa (Table 3).

The quality of modern contraceptive varies with the management authority of Health facilities; NGOs had the highest quality (73.24%) with higher and private facilities scored the lowest quality (58.11%). Moreover, according to areas of health facilities, facilities in urban areas had higher quality (68.54%) (Fig 2).

The quality of modern contraceptives also varies with the type of health facilities; hospitals had the highest quality (72%), whereas hospitals in Harari had the best quality 80% whereas hospitals in Gambela had a lower quality (64%) (Table 4).

### Effective coverage

The national effective coverage of modern contraceptive services was 20% (95% CI: 19.87, 21.82), which was notably lower than the crude coverage of 30% (95% CI: 28.93, 31.62), resulting in a gap of 10% points. The highest effective coverage of modern contraceptive services was observed in SNNP at 25% and Amhara at 24%, with gaps between crude and effective coverage of 12 and 11 percentage points, respectively. Conversely, the lowest effective coverage was

**Table 1. Socioeconomic and demographic characteristics of participants in Ethiopia, EMDHS 2019 (n = 7618).**

| Variables | Categories | Number | Frequency% |
|---|---|---|---|
| **Age** | 15–24 | 3,219 | 42.25 |
| | 25–35 | 2,743 | 36.01 |
| | 36–49 | 1,656 | 21.74 |
| **Marital status** | Married | 4,875 | 63.99 |
| | Unmarried | 2,743 | 36.01 |
| **Residence** | Rural | 5,171 | 67.88 |
| | Urban | 2,447 | 32.12 |
| **Religion** | Orthodox | 2,915 | 38.27 |
| | Muslim | 2,342 | 30.75 |
| | Protestant | 2,238 | 29.37 |
| | *Others | 123 | 1.61 |
| **Region** | Oromia | 3070 | 40.3 |
| | Amhara | 1915 | 25.14 |
| | SNNP | 1574 | 20.66 |
| | Addis Abeba | 411 | 5.4 |
| | Somali | 362 | 4.75 |
| | Benishangul-Gumuz | 90 | 1.19 |
| | Afar | 76 | 0.99 |
| | Dire Dawa | 59 | 0.77 |
| | Gambela | 37 | 0.49 |
| | Harari | 24 | 0.31 |
| **Education** | No education | 3,077 | 40.39 |
| | Primary education | 3,255 | 42.73 |
| | Secondary education | 905 | 11.89 |
| | Higher | 381 | 5 |
| **Wealth** | Poorest | 1,182 | 15.51 |
| | Poorer | 1,416 | 18.59 |
| | Middle | 1,468 | 19.27 |
| | Richer | 1,686 | 22.13 |
| | Richest | 1,866 | 24.5 |

*SNNP: Southern Nations, Nationalities and Peoples; *Catholic, traditional and others; EMDHS: Ethiopia mini demographic and health survey*

**Table 2. Health facilities characteristics ESPA 2021-2022 (n = 648).**

| Variable | Categories | Frequency | Percentage |
|---|---|---|---|
| **Facility type** | Hospital | 285 | 43.98 |
| | Health centre | 209 | 32.25 |
| | Health post | 64 | 9.88 |
| | Clinic | 90 | 13.89 |
| **Area** | Urban | 374 | 57.72 |
| | Rural | 274 | 42.28 |
| **Managing authority** | Government | 534 | 82.41 |
| | Private | 100 | 15.43 |
| | NGO | 14 | 2.16 |

*NGO: Non-Governmental Organisation; ESPA: Ethiopia service provision assessment*

**Table 3. Quality of family planning services in ESPA 2021-2022(n = 648).**

| Region | Facility | Structural (95% CI) | Process (95% CI) | Outcome (95% CI) | Quality (95% CI) |
|---|---|---|---|---|---|
| **Addis Ababa** | 43 | 77.53 (73.20, 81.86) | 60.68 (57.12, 64.24) | 90.97 (86.96, 94.98) | 73.70 (71.00, 76.40) |
| **Afar** | 32 | 57.71 (50.40, 65.02) | 47.95 (43.44, 52.46) | 90.07 (85.58, 94.56) | 61.30 (57.50, 65.10) |
| **Amhara** | 117 | 71.51 (68.56, 74.46) | 53.38 (51.24, 55.52) | 92.50 (89.56, 95.44) | 68.90 (67.10, 70.70) |
| **Benishangul-Gumuz** | 29 | 65.61 (57.20, 74.02) | 48.37 (44.64, 52.10) | 97.96 (94.94, 100.98) | 66.10 (61.90, 70.20) |
| **Dire Dawa** | 20 | 71.75 (64.80, 78.70) | 54.58 (48.74, 60.42) | 94.11 (87.06, 101.16) | 69.90 (66.30, 73.50) |
| **Gambela** | 51 | 55.73 (50.13, 61.33) | 48.73 (45.61, 51.85) | 81.08 (75.84, 86.32) | 58.80 (56.20, 61.40) |
| **Harari** | 11 | 66.50 (54.72, 78.28) | 60.53 (49.18, 71.88) | 92.51 (86.68, 98.34) | 70.20 (61.80, 78.60) |
| **Oromia** | 164 | 65.86 (63.07, 68.65) | 53.60 (52.08, 55.12) | 91.13 (89.02, 93.24) | 66.70 (65.20, 68.30) |
| **SNNP** | 155 | 66.02 (63.57, 68.47) | 56.63 (55.20, 58.06) | 90.54 (88.19, 92.89) | 67.90 (66.50, 69.30) |
| **Somali** | 26 | 63.34 (54.14, 72.54) | 48.02 (42.73, 53.31) | 76.92 (68.23, 85.61) | 60.20 (54.90, 65.40) |
| **National Level** | 648 | 66.57 (63.65, 69.49) | 53.77 (52.77, 54.77) | 90.24 (88.62, 91.86) | 66.80 (66.00, 67.60) |

*SNNP: Southern Nations, Nationalities and Peoples; CI: Confidence Interval; ESPA: Ethiopia service provision assessment*

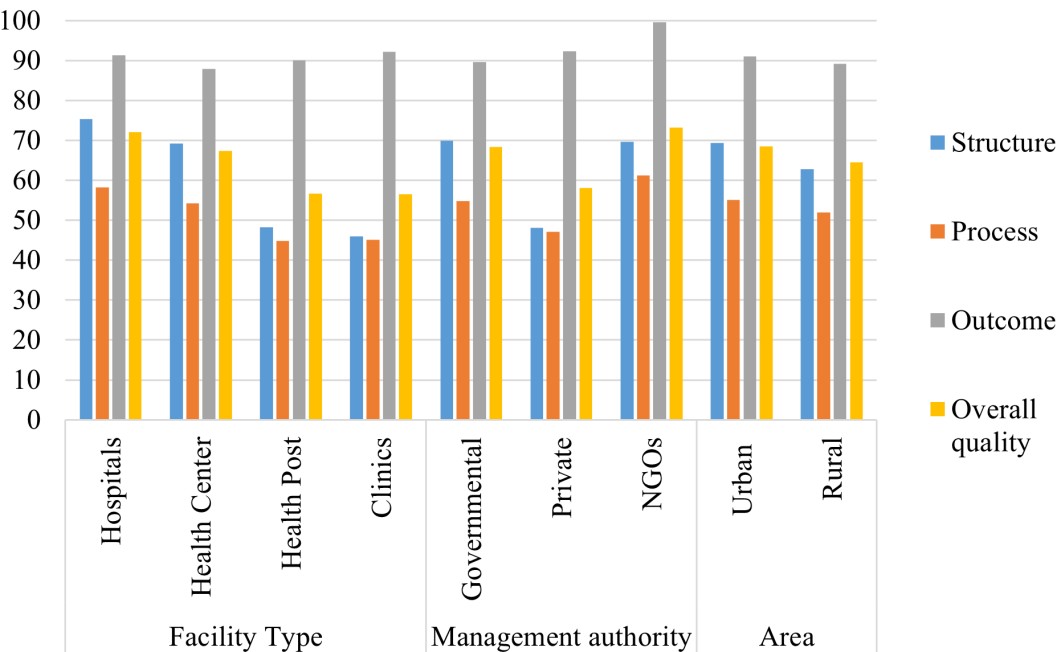

**Fig 2. Quality score of modern contraceptive services by facility characteristic in ESPA 2021-2022.**

recorded in Somali at 5% with coverage gaps of 3% points (Table 5). Effective coverage results with administrative linkage were reported (S2 Table).

## Spatial distribution and hotspot analysis of effective coverage of modern contraceptive services

The effective coverage of modern contraceptive services showed a significant positive spatial autocorrelation in Ethiopia, with the Global Moran's index value of 1.92 (p-value < 0.001) (Fig 3).

The spatial distribution analysis showed central Amhara, western Amhara central Oromia, western part of South West Ethiopia, Sidama and the northern part of SNNP had higher effective coverage of modern contraceptive services (Fig 4).

**Table 4. Quality of health facilities by region, ESPA 2021-2022.**

| Region (n) | Quality (95% CI) | | | | |
|---|---|---|---|---|---|
| | H | HC | HP | C | Subtotal |
| **Addis Ababa (43)** | 76.58 (73.50, 79.66) | 73.98 (71.29, 76.67) | N/A | 53.39 (19.19, 87.59) | 73.7 (71.0, 76.4) |
| **Afar (32)** | 66.97 (59.82, 74.12) | 63.43 (58.67, 68.19) | 48.60 (33.51, 63.69) | 59.63 (49.13, 70.13) | 61.3 (57.5, 65.1) |
| **Amhara (117)** | 71.99 (70.12, 73.86) | 69.71 (66.56, 72.86) | 56.68 (48.28, 65.08) | 58.18 (51.14, 65.22) | 68.9 (67.1, 70.7) |
| **Benishangul-Gumuz (29)** | 78.46 (69.02, 87.90) | 69.47 (65.56, 73.38) | 58.33 (54.29, 62.37) | 53.00 (44.67, 61.33) | 66.1 (61.9, 70.2) |
| **Dire Dawa (20)** | 69.28 (52.61, 85.95) | 71.98 (69.39, 74.57) | 56.24 (29.83, 82.65) | 71.21 (34.65, 107.77) | 69.9 (66.3, 73.5) |
| **Gambela (51)** | 63.64 (57.73, 69.55) | 64.62 (60.81, 68.43) | 52.02 (46.48, 57.56) | 55.09 (51.03, 59.15) | 58.8 (56.2, 61.4) |
| **Harari (11)** | 80.32 (66.27, 94.37) | 72.10 (67.29, 76.91) | 56.78 (14.85, 98.71) | N/A | 70.2 (61.8, 78.6) |
| **Oromia (164)** | 71.92 (70.70, 73.14) | 65.12 (62.36, 67.88) | 56.53 (51.81, 61.25) | 54.31 (49.28, 59.34) | 66.7 (65.2, 68.3) |
| **SNNP (155)** | 71.12 (69.35, 72.89) | 66.89 (64.80, 68.98) | 61.37 (57.26, 65.48) | 60.19 (52.79, 67.59) | 67.9 (66.5, 69.3) |
| **Somali (26)** | 73.02 (68.25, 77.79) | 54.32 (47.71, 60.93) | 50.69 (−81.66, 83.04) | 52.85 (39.37, 66.33) | 60.2 (54.9, 65.4) |
| **National Level (648)** | 71.98 (71.15, 72.81) | 67.35 (66.20, 68.50) | 56.71 (54.44, 58.98) | 56.54 (54.24, 58.84) | 66.8 (66.0, 67.6) |

SNNP: Southern Nations, Nationalities and Peoples; H: Hospitals; HC: Health centre; HP: Health post; C: Clinic; N/A: Not available; n: number of health facilities; ESPA: Ethiopia service provision assessment

**Table 5. Effective coverage of modern contraceptive services in Ethiopia.**

| Region | Crude coverage (95% CI) | EC % (95% CI) | CC-EC (% point) |
|---|---|---|---|
| **Addis Ababa** | 28.18 (23.79, 32.56) | 20.84 (19.87, 21.82) | 7.34 |
| **Afar** | 16.56 (10.83, 22.30) | 10.09 (6.99, 13.19) | 6.47 |
| **Amhara** | 35.64 (32.81, 38.48) | 24.44 (22.48, 26.44) | 11.20 |
| **Benishangul-Gumuz** | 28.68 (24.54, 32.81) | 18.89 (16.02, 21.76) | 9.79 |
| **Dire Dawa** | 20.80 (14.41, 27.18) | 14.61 (13.39, 15.83) | 6.19 |
| **Gambela** | 26.87 (19.33, 34.42) | 16.77 (12.78, 20.76) | 10.10 |
| **Harari** | 22.50 (14.74, 30.27) | 14.78 (11.89, 17.67) | 7.72 |
| **Oromia** | 30.76 (28.41, 33.10) | 20.51 (18.87, 22.15) | 10.25 |
| **SNNP** | 36.10 (33.60, 38.59) | 24.54 (22.80, 26.29) | 11.56 |
| **Somali** | 7.96 (4.36, 11.57) | 4.79 (2.56, 7.03) | 3.17 |
| **National level** | 30.27 (28.93, 31.62) | 20.41 (19.54, 21.28) | 9.86 |

SNNP: Southern Nations, Nationalities and Peoples; EC: Effective coverage; CC: Crude coverage

There were hot and cold spots areas for effective coverage of modern contraceptive services in Ethiopia. Addis Ababa, western Amhara, northeast Benishangul-Gumuz, eastern part of South West Ethiopia, north SNNP, Sidama, and northern and central Oromia had hotspots for effective coverage of modern contraceptive services. On the other hand, Afar, western Gambela, western Oromia, western Benishangul-Gumuz, Dire Dawa, Harari and Somali had cold spots for effective coverage of modern contraceptive services (Fig 5).

## Spatial interpolation of modern contraceptive services in Ethiopia

The spatial prediction of effective coverage of modern contraceptive services showed a high coverage mainly in northwest Amhara, central Amhara, central Oromia, southern Gambela, Sidama, western part of SNNP, and western and southern part of South West Ethiopia. There was low effective coverage of modern contraceptive services in Somali, northern Afar, southern Afar and western Gambela. The predicted low effective coverage of modern contraceptive services was shown in red colour (Fig 6).

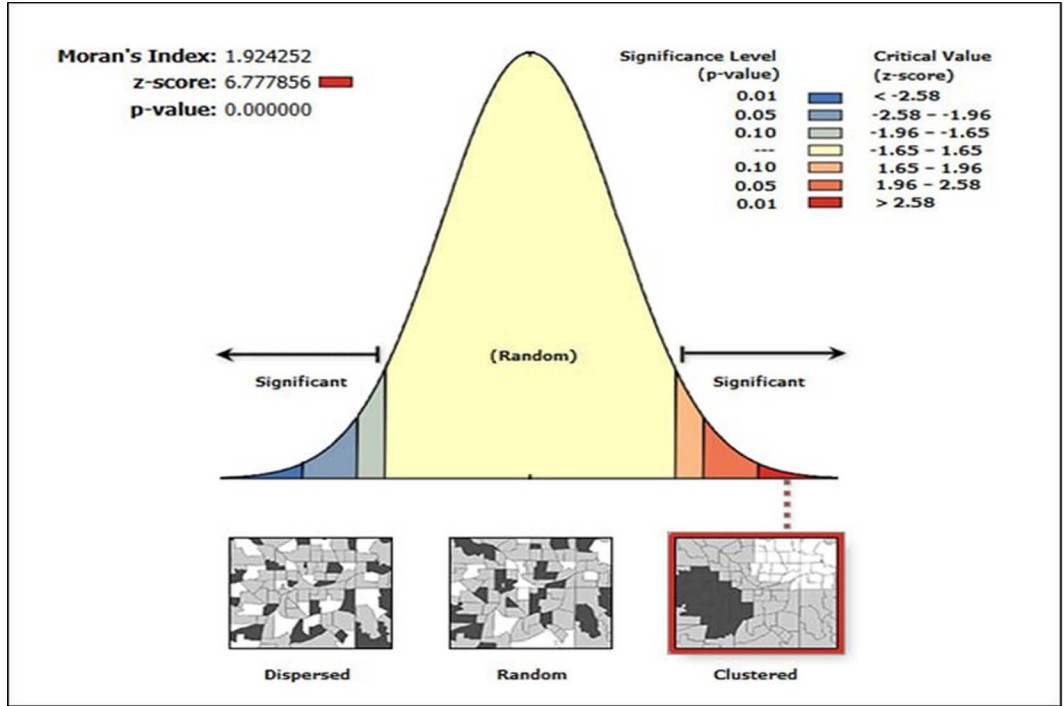

**Fig 3. Spatial autocorrelation analysis of effective coverage of modern contraceptive services in Ethiopia.**

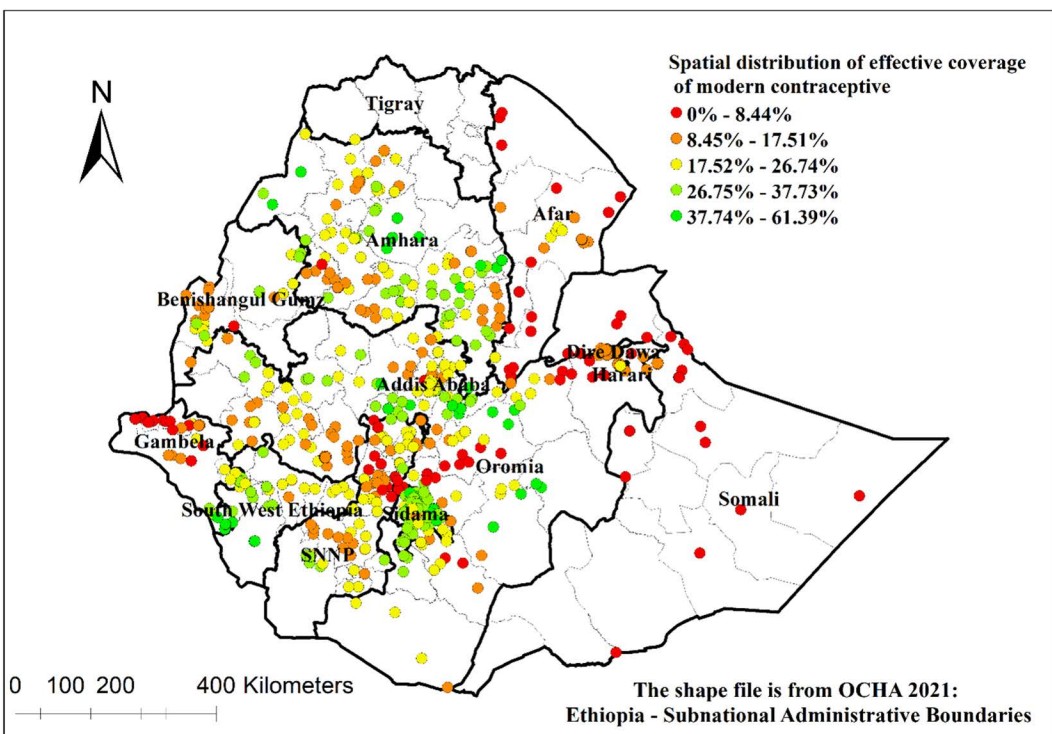

**Fig 4. Spatial distribution of effective coverage of modern contraceptive services in Ethiopia.**

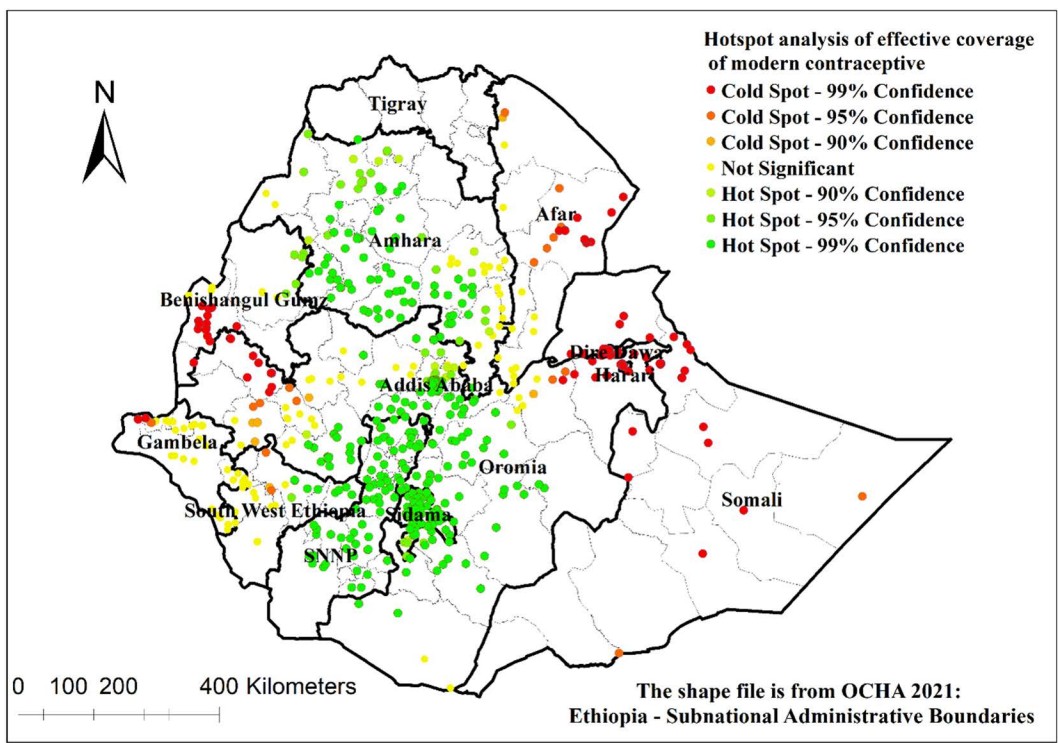

**Fig 5. Hotspot analysis of effective coverage of modern contraceptive services in Ethiopia.**

## Discussion

This study revealed the spatial distribution of effective coverage of modern contraceptive services in Ethiopia using the three components of quality measurements. The effective coverage of modern contraceptive services was 20%, with a large variation in different regions. A previous similar study in Ethiopia also found that the effective coverage (22%) of family planning services was low compared to its crude coverage (66%), with significant geographical variation among health facilities [24]. A study conducted in different African countries also reported that the effective coverage of family planning was 16% in Haiti and 19% in Senegal [18]. This might be due to the low quality of health facility infrastructure, health services, supplies and experienced health providers. This study found Afar, western Gambela, western Oromia, western Benishangul-Gumuz, Dire Dawa, Harari and Somali had low effective coverage. This was consistent with a previous study conducted in Ethiopia [44]. A possible reason might be the availability of health services and accessibility of health services in peripheral regions, which results in missing the opportunity to give quality family planning [45,46].

Improved quality of family planning services will increase the utilisation of women [47], which as a result increases the effective coverage of family planning services. The overall quality score of modern contraception services in this study was 67%, indicating a moderate level of service quality. Similarly, research in Kenya found that the quality of family planning services was 47%, which was lower than the findings of this study [18]. These variations may be due to differences in health system capacity, service delivery models, and accessibility of contraceptive services across different settings.

The quality of modern contraception services varied across regions, ranging from 73.7% in Addis Ababa to 59% in Gambela, indicating regional disparities. This finding was consistent with a previous study that reported regional differences in service quality, with scores ranging from 28% in Amhara to 49% in Addis Ababa [48]. Comparatively, a study conducted in Dire Dawa reported that only 37% of women received good-quality family planning services, highlighting a

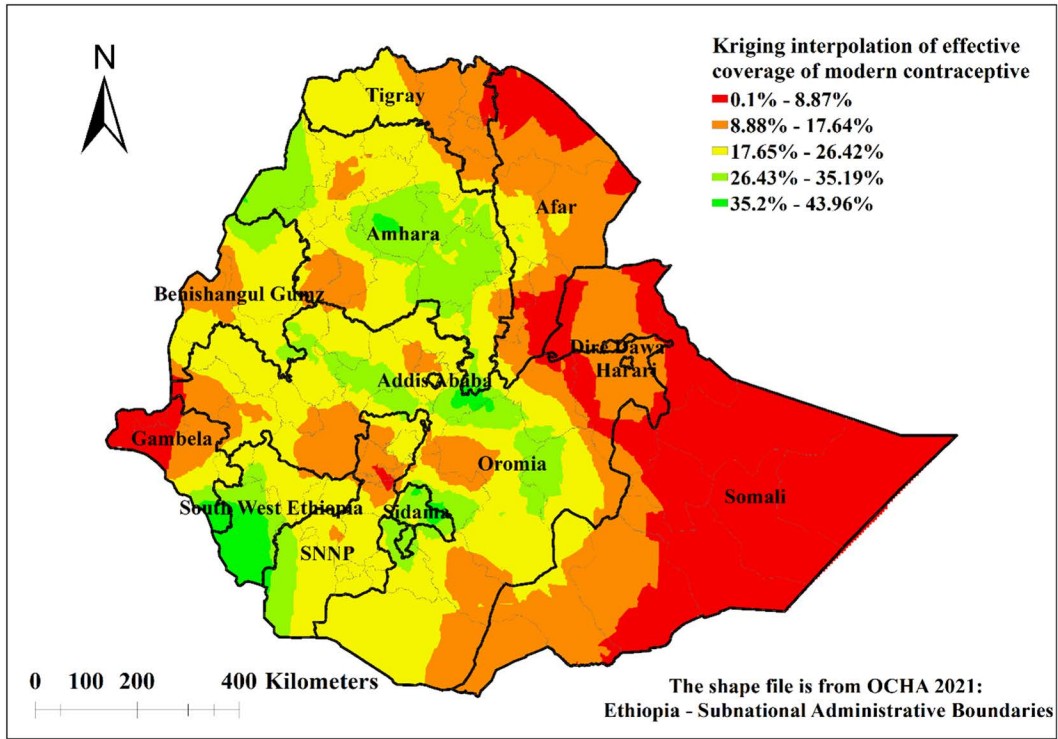

**Fig 6. Spatial interpolation of effective coverage of modern contraceptive services in Ethiopia.**

significant gap in service provision [22]. Our study further revealed that urban health facilities had a higher quality score (69%). This finding aligns with previous research conducted in Ethiopia, which reported that facilities in urban areas (38%) were more likely to provide better-quality family planning services than those in rural settings (34%) [24]. These disparities may be due to the higher employment of experienced personnel in urban areas, better infrastructure, and easier access to health supplies [49]. Additionally, higher literacy levels enhance client-provider interactions, while greater economic stability, education, and healthcare access further improve service quality and utilisation in urban settings [50].

Our study found that government facilities provided higher-quality modern contraceptive services (68%) compared to private facilities (58%). This finding was consistent with previous research conducted in Ethiopia, which reported that public facilities offered slightly better service quality than private facilities, with a quality score of 35% in public facilities compared to 33% in private ones [24]. One possible explanation for the higher quality of services in government facilities was that they were often better integrated into national healthcare programs, receiving continuous training, resources, and monitoring to ensure adherence to service quality standards [51,52].

The study used women's satisfaction to measure the quality of outcome components. The outcome component had the highest quality score, 90%. This was consistent with a study in Jimma, Southwest Ethiopia (94%) [53]. Moreover, the finding was higher than a study conducted in Mozambique (85%) (47) and Nigeria (81%) [54]. The structural component had the second score (67%), ranging from 56% in Gambela to 78% in Addis Ababa. A study conducted in Ethiopia showed that the readiness of health facilities for family planning was 78%, ranging from 74% in Afar to 90% in Harari [55]. Lastly, the process component had the lowest quality score (54%), ranging from 48% in Afar to 61% in Addis Ababa. The lower score for the process component may suggest gaps in service delivery, provider-client interactions, or adherence to clinical

guidelines. Since the process component reflects the actual implementation of healthcare services, targeted interventions to enhance provider training, counselling practices, and services integration could help improve overall quality [56].

**Strengths and limitations of the study**

This study had numerous strengths, including using the DHS dataset, which had a high sample size, and the most recent EMDHS dataset, 2019. However, the study had some limitations. The first limitation of the study was that we did not exclude the use of contraceptives from other sources, such as pharmacies, because EMDHS 2019 did not contain any observations regarding the source of contraceptives. The other limitation was that the Tigray region was not included, as the ESPA does not have a record for facilities in Tigray, and only health facilities that had available exit interview records related to family planning services were included. A limitation of our matching methods was that they do not account for geographic barriers, such as mountains or the absence of roads, which can affect movement or access between locations. The Euclidean distance-based approach assumes straight-line proximity, but this may not accurately reflect the true travel distances or accessibility in real-world settings. As a result, the model may overestimate the closeness of certain locations or understate the challenges posed by physical barriers to movement, which could influence the study outcomes.

## Conclusions

There is significant spatial variation in the effective coverage of modern contraceptive services across Ethiopia. Afar, the western part of Gambela, western Oromia, western Benishangul Gumuz, Dire Dawa, Harari, and Somali, remain significantly underserved. To promote equitable access, policymakers should prioritise targeted interventions that strengthen healthcare infrastructure, improve service delivery, and address regional disparities in the availability and quality of modern contraceptive services.

## Supporting information

**S1 Table.** Quality indicator variables for modern contraceptive services (SPA 2021–2022).
(DOCX)

**S2 Table.** Effective coverage of modern contraceptive services in Ethiopia 2019 EMDHS, with administrative data linkage.
(DOCX)

## Author contributions

**Conceptualization:** Samrawit Birhanu Alemu.

**Formal analysis:** Samrawit Birhanu Alemu, Molla Yigzaw Birhanu, Mekdes Tamiru Yizengaw, Melaku Birhanu Alemu.

**Methodology:** Samrawit Birhanu Alemu, Molla Yigzaw Birhanu, Mekdes Tamiru Yizengaw, Melaku Birhanu Alemu.

**Writing – original draft:** Samrawit Birhanu Alemu, Molla Yigzaw Birhanu, Mekdes Tamiru Yizengaw, Melaku Birhanu Alemu.

**Writing – review & editing:** Melaku Birhanu Alemu.

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
