## [Decision Letter · Decision Letter 0]

PONE-D-24-60655Spatial distribution of effective coverage of modern contraceptive service in EthiopiaPLOS ONE

Dear Dr. Alemu,

Thank you for submitting your manuscript to PLOS ONE. After careful consideration, we feel that it has merit but does not fully meet PLOS ONE’s publication criteria as it currently stands. Therefore, we invite you to submit a revised version of the manuscript that addresses the points raised during the review process.

We look forward to receiving your revised manuscript.

Kind regards,

Yitagesu Habtu Aweke, Ph.D

Academic Editor

PLOS ONE

Journal Requirements:

2. In the online submission form, you indicated that the Ethiopian demographic health survey (DHS) dataset is available to the general public upon request from the Measure DHS website http://www.measuredhs.com.

3. We note that Figures 4, 5 and 6 in your submission contain [map/satellite] images which may be copyrighted. All PLOS content is published under the Creative Commons Attribution License (CC BY 4.0), which means that the manuscript, images, and Supporting Information files will be freely available online, and any third party is permitted to access, download, copy, distribute, and use these materials in any way, even commercially, with proper attribution. For these reasons, we cannot publish previously copyrighted maps or satellite images created using proprietary data, such as Google software (Google Maps, Street View, and Earth). For more information, see our copyright guidelines: http://journals.plos.org/plosone/s/licenses-and-copyright.

a. You may seek permission from the original copyright holder of Figures 4, 5 and 6 to publish the content specifically under the CC BY 4.0 license.  

Reviewers' comments:

Reviewer's Responses to Questions

**Comments to the Author**

1. Is the manuscript technically sound, and do the data support the conclusions?

Reviewer #1: Yes

Reviewer #2: Partly

2. Has the statistical analysis been performed appropriately and rigorously? 

Reviewer #1: No

Reviewer #2: No

3. Have the authors made all data underlying the findings in their manuscript fully available?

Reviewer #1: Yes

Reviewer #2: Yes

4. Is the manuscript presented in an intelligible fashion and written in standard English?

Reviewer #1: No

Reviewer #2: No

5. Review Comments to the Author

Reviewer #1: Abstract

Avoid using acronyms at this section

Please include the clear gap beyond the coverage.

It would be better if you included keywords

Introduction

• There is an idea stated “Therefore, this indicates the prevalence of unmet need for modern contraceptives among married or in-union women is around 10% [7].” in paragraph 2.

It is not clear how the unmet need was calculated

And even it is better to focus on the study area like sub-Saharan, Ethiopia

• Try to include the figures how the poor quality affects the quality of health.

• Try to include how poor quality related with effective coverage.

• Despite it was not adjusted with quality of service, I recommend you to include the figure of the coverage.

• There are acronyms without any elaborations in their first stand

Methods

1. Measurement

How did you measure the frequency and or duration of contraceptive use? Is it a history/ever used or a current use?

How the quality of measurement was determined? When did you say poor quality or good quality?

2. There are missed components that had to be reported despite it was a secondary data analysis.

What was the study design?

What was the study population?

What was the sampling technique?

How did you assure the data quality?

Ethical consideration

3. Data analysis

Spatial autocorrelation, spatial interpolation, and hotspot analysis had to be written separately as subheadings.

What model was used? How the model chosen?

How the samples weighted?

What way of measurement was used to measure variations between regions?

Result

Your text report regarding “Age” is inappropriate. It says 42.25% below 42 and 21.74% above 36: What does it mean? Does it mean 0-24 and 36 to ……….?

The total number of participants (n) is not the same as that you stated in some variables. For example marital status (n=7619).

Why this happened?

Better to include “total” as a category for each variable

Categories of some variables are not logical sequenced. For example region

I recommend you to sequence variables in logical manner using their frequency either ascending or descending

Some graphs lacks the minimum standard that should contain like Legend, caption

The footnote for table 5 is not written in a recommended way.

Better to state it under the table in text form with a specific note (to attract the reader)

Discussion

In some points you tried to discuss other’s study rather than yours which is not a recommended scientific way of writing.

In some points you said various studies/many studies/different studies but you cited only a single reference.

In some points you said other’s finding low coverage and or quality.

Why preferred to report or compare and contrast in this way?

Why didn’t you compare and contrast in a quantitative way (or why didn’t you show their difference and or similarity rather than concluding others?) which is measurable to judge for the readers.

It even difficult to conclude other’s finding

You discussed with a study done on counseling quality which is not similar with yours.

There is an idea which says “Moreover, Facilities in urban areas had higher quality (68.54%).”

Is it your finding or others’?

Whether it is your or others’, it is not a recommended scientific ways of writing.

Conclusion

“The study found that, Ethiopia had a poor effective coverage of modern contraceptives services”

This idea is a copy paste of the first sentence.

“Somalia and Afar regions had less effective coverage of modern contraceptive service.”

This is also repeated

“Policymakers and responsible bodies should focus on improving the quality of the facilities in rural areas and the facilities in Gambela, Somali and Afar”

This is also repeated

Reviewer #2: Dear author,

Thanks for your kind attention to the different factors for client satisfaction. I think you need to update some of the definitions to well accepted ones. Also you need to provide definition of some words like poor, low and high in your manuscript.

While an indicator defined low in your manuscript, could be moderate or even high in somewhere else.

I provided my comments on the text and will attach two files to this review.

6. PLOS authors have the option to publish the peer review history of their article (what does this mean? ). If published, this will include your full peer review and any attached files.

**Do you want your identity to be public for this peer review?** For information about this choice, including consent withdrawal, please see our Privacy Policy .

Reviewer #1: **Yes: ** Yitayal Ayalew Goshu

Reviewer #2: **Yes: ** Mohammad Eslami

---

## [Author Response · Author response to Decision Letter 1]

6 Apr 2025

Thank you to the reviewers and editor for taking the time to review our manuscript. We have addressed the comments and revised the manuscript accordingly. We have uploaded the revised version in both clean and track-changes formats, along with a point-by-point response.

---

## [Decision Letter · Decision Letter 1]

PONE-D-24-60655R1Spatial distribution of effective coverage of modern contraceptive service in EthiopiaPLOS ONE

Dear Dr. Alemu,

Thank you for submitting your manuscript to PLOS ONE. After careful consideration, we feel that it has merit but does not fully meet PLOS ONE’s publication criteria as it currently stands. Therefore, we invite you to submit a revised version of the manuscript that addresses the points raised during the review process.

We look forward to receiving your revised manuscript.

Kind regards,

Yitagesu Habtu Aweke, Ph.D

Academic Editor

PLOS ONE

Additional Editor Comments :

Please check writing issues in all of your sections or subsection of your manuscript, specifically in your discussion section. Eg. punctuation marks in your abstract section, methods subsection line 4^th^ line, “… 1.92 with (p< 0.001)”, results subsection “…contraceptive” or contraceptives? of your abstract …Check that all of your references are according to PLoS citations, and referencesCheck the journal’s outstanding criteria from the authors guideline. E.g. “**Patient and public involvement?” **Check that your figures are a per the submission guidelines  

Reviewers' comments:

Reviewer's Responses to Questions

**Comments to the Author**

1. If the authors have adequately addressed your comments raised in a previous round of review and you feel that this manuscript is now acceptable for publication, you may indicate that here to bypass the “Comments to the Author” section, enter your conflict of interest statement in the “Confidential to Editor” section, and submit your "Accept" recommendation.

Reviewer #2: All comments have been addressed

2. Is the manuscript technically sound, and do the data support the conclusions?

Reviewer #2: Partly

3. Has the statistical analysis been performed appropriately and rigorously? 

Reviewer #2: I Don't Know

4. Have the authors made all data underlying the findings in their manuscript fully available?

Reviewer #2: Yes

5. Is the manuscript presented in an intelligible fashion and written in standard English?

Reviewer #2: Yes

6. Review Comments to the Author

Reviewer #2: Thanks for addressing the comment. Unfortunately I couldn't find some revisions in the text. In your responses, you mentioned the lines number which there is no line number in the revised version.

It will be useful if you provide the exact revised line numbers for each response also, use a version with line numbers which make finding the revisions easier.

7. PLOS authors have the option to publish the peer review history of their article (what does this mean? ). If published, this will include your full peer review and any attached files.

**Do you want your identity to be public for this peer review?** For information about this choice, including consent withdrawal, please see our Privacy Policy .

Reviewer #2: **Yes: ** Mohammad Eslami

---

## [Author Response · Author response to Decision Letter 2]

26 May 2025

We sincerely thank the reviewers for their insightful feedback and constructive comments. We apologize for the oversight regarding the missing line numbers in the revised manuscript. We have updated each response to include the exact line numbers where the corresponding revisions can be found. Please refer to the newly submitted version with line numbers for easy cross-referencing.

---

## [Editor Report · Decision Letter 2]

PONE-D-24-60655R2

Mapping the effective coverage of modern contraceptive services in Ethiopia

PLOS ONE

Dear Dr. Alemu,

Thank you for submitting your revised manuscript to PLOS ONE. After careful consideration, we feel that it has merit and about to meet PLOS ONE’s publication criteria as it currently stands. Therefore,  we invite you to submit a revised version of the manuscript that addresses a very minor point raised during the review process.

We look forward to receiving your revised manuscript.

Kind regards,

Yitagesu Habtu Aweke, Ph.D

Academic Editor

PLOS ONE

Additional Editor Comments:

 "Patient and public involvement" is not typically part of the journal's criteria for manuscripts of this type, as the study did not include active participation from patients or the public, despite the obvious relevance of its findings to them. 

---

## [Author Response · Author response to Decision Letter 3]

3 Jun 2025

We appreciate the thoughtful comments and suggestions provided by you and the reviewers, which have helped us to improve the clarity and quality of our work.

---

## [Editor Report · Decision Letter 3]

Mapping the effective coverage of modern contraceptive services in Ethiopia

PONE-D-24-60655R3

Dear Dr. Samrawit Alemu,

We’re pleased to inform you that your manuscript has been judged scientifically suitable for publication and will be formally accepted for publication once it meets all outstanding technical requirements.

Kind regards,

Yitagesu Habtu Aweke, Ph.D

Academic Editor

PLOS ONE

---

## [Editor Report · Acceptance letter]

PONE-D-24-60655R3

PLOS ONE

Dear Dr. Birhanu Alemu,

I'm pleased to inform you that your manuscript has been deemed suitable for publication in PLOS ONE. Congratulations! Your manuscript is now being handed over to our production team.

Kind regards,

on behalf of

PhD Candidate Yitagesu Habtu Aweke

Academic Editor

PLOS ONE